# Transmembrane protein CD69 acts as an S1PR1 agonist

Hongwen Chen[1], Yu Qin[1], Marissa Chou[2], Jason G Cyster[2,3]*, Xiaochun Li[1,4]*

[1]Department of Molecular Genetics, The University of Texas Southwestern Medical Center, Dallas, United States; [2]Department of Microbiology and Immunology, University of California, San Francisco, San Francisco, United States; [3]Howard Hughes Medical Institute, University of California, San Francisco, San Francisco, United States; [4]Department of Biophysics, The University of Texas Southwestern Medical Center, Dallas, United States

**Abstract** The activation of Sphingosine-1-phosphate receptor 1 (S1PR1) by S1P promotes lymphocyte egress from lymphoid organs, a process critical for immune surveillance and T cell effector activity. Multiple drugs that inhibit S1PR1 function are in use clinically for the treatment of autoimmune diseases. Cluster of Differentiation 69 (CD69) is an endogenous negative regulator of lymphocyte egress that interacts with S1PR1 in cis to facilitate internalization and degradation of the receptor. The mechanism by which CD69 causes S1PR1 internalization has been unclear. Moreover, although there are numerous class A GPCR structures determined with different small molecule agonists bound, it remains unknown whether a transmembrane protein per se can act as a class A GPCR agonist. Here, we present the cryo-EM structure of CD69-bound S1PR1 coupled to the heterotrimeric G$_i$ complex. The transmembrane helix (TM) of one protomer of CD69 homodimer contacts the S1PR1-TM4. This interaction allosterically induces the movement of S1PR1-TMs 5–6, directly activating the receptor to engage the heterotrimeric G$_i$. Mutations in key residues at the interface affect the interactions between CD69 and S1PR1, as well as reduce the receptor internalization. Thus, our structural findings along with functional analyses demonstrate that CD69 acts in cis as a protein agonist of S1PR1, thereby promoting G$_i$-dependent S1PR1 internalization, loss of S1P gradient sensing, and inhibition of lymphocyte egress.

*For correspondence:
jason.cyster@ucsf.edu (JGC);
xiaochun.li@utsouthwestern.
edu (XL)

Competing interest: The authors declare that no competing interests exist.

## Editor's evaluation

This important study provides unprecedented molecular insight into the activation and internalization of an important cell surface receptor induced by another membrane protein. The data supporting the conclusions are compelling, which include rigorous electron microscopy analysis, and biochemical and cell-based functional assays. The findings here not only reveal important mechanisms of S1P GPCR regulation, but also have implications for other fields such as receptor pharmacology and immunity.

## Introduction

Sphingosine-1-phosphate (S1P) plays an essential role in the immune system by promoting the egress of lymphocytes from lymphoid organs into blood and lymph via a direct interaction with one of its five cognate G protein–coupled receptors, S1PR1 (*Baeyens and Schwab, 2020*; *Cartier and Hla, 2019*; *Cyster and Schwab, 2012*; *Pappu et al., 2007*; *Rosen et al., 2013*; *Spiegel and Milstien, 2003*). After egressing from spleen, lymph nodes, or mucosal lymphoid tissues, T and B lymphocytes travel to other lymphoid organs in a cycle of continual pathogen surveillance. When an infection occurs,

there is a temporary shutdown of lymphocyte egress from the responding lymphoid organ(s) and this enables increased accumulation of lymphocytes and enhances the immune response (*Cyster and Schwab, 2012*). Egress shutdown is mediated by type I interferon (IFN) inducing lymphocyte CD69 expression. CD69, a type II transmembrane C-type lectin protein, intrinsically inhibits the function of S1PR1 in T and B cells (*Shiow et al., 2006*). CD69 also regulates T cell egress from the thymus (*Nakayama et al., 2002*; *Zachariah and Cyster, 2010*). A disulfide-bond in the extracellular domain links CD69 as a homodimer (*Ziegler et al., 1994*). Biochemical studies demonstrated that CD69 may associate with S1PR1 through interactions between their transmembrane domains (TMs) to facilitate S1PR1 internalization and degradation (*Bankovich et al., 2010*). Unlike S1P, CD69 has been shown to bind S1PR1 but not the other S1PRs (*Bankovich et al., 2010*; *Jenne et al., 2009*; *Shiow et al., 2006*). However, the mechanism of CD69-induced S1PR1 internalization and thus functional inactivation has been unclear.

Importantly, several S1PR1 modulators (e.g. Fingolimod, also known as FTY720, Siponimod, Ozanimod, and Etrasimod), have been approved for treating the autoimmune diseases multiple sclerosis and ulcerative colitis (*Brinkmann et al., 2010*; *Chun et al., 2021*; *Dal Buono et al., 2022*; *Kappos et al., 2010*). These immunosuppressants are believed to act by inhibiting S1PR1 function and thereby preventing autoimmune effector lymphocytes exiting lymphoid organs, blocking the autoimmune attack. Either sphingosine or FTY720 is metabolically catalyzed by two intracellular sphingosine kinases into the phosphorylated form (S1P or FTY720-P) and then exported to the extracellular space via S1P transporters (*Baeyens and Schwab, 2020*; *Spiegel et al., 2019*). There, S1P binds to its receptors for initiation of the signal while FTY720-P activates the S1PR1 but causes a persistent internalization and degradation of S1PR1 to attenuate the signal (*Brinkmann et al., 2010*).

Recently, cryogenic electron microscopy (cryo-EM) structures of S1PR1 complexed with different small molecule ligands have been determined (*Liu et al., 2022*; *Xu et al., 2022*; *Yu et al., 2022*; *Yuan et al., 2021*). These findings reveal a mechanism of how S1PR1 engages its endogenous ligand S1P and its modulators to adopt the active conformation for recruiting the heterotrimeric $G_i$ protein. The previously determined crystal structure of antagonist ML056-bound S1PR1 reveals its inactive state (*Hanson et al., 2012*). However, the molecular mechanism remains unknown of how CD69 binds to S1PR1 to trigger its internalization. Therefore, structural study on the S1PR1-CD69 complex will provide molecular insights into the CD69-mediated functional inhibition of S1PR1 and reveal how a class A GPCR can be regulated by a transmembrane protein modulator. In this manuscript, we determined the structure of CD69-bound S1PR1 coupled to $G_{\alpha i \beta 1 \gamma 2}$ heterotrimer by cryo-EM at 3.15 Å resolution. Our findings reveal that TM of CD69 contacts TM4 of S1PR1 to activate the receptor allowing it to engage the α5 helix of $G_{\alpha i}$ in the absence of S1P ligand, thereby disrupting the receptor's egress-promoting function.

## Results

Since serum contains an abundance of lipids including S1P, we expressed human S1PR1 or CD69 in HEK293 cells cultured in a medium with lipid-deficient serum. Then, we purified human S1PR1 protein alone to validate its activation in the presence of S1P using the GTPase-Glo assay (*Figure 1A*). We then tested the effect on S1PR1 of adding the CD69 homodimer in the absence of S1P. Remarkably, addition of CD69 alone caused a similar amount of $G_i$ activation as addition of S1P indicating that CD69 functions as a protein agonist of S1PR1 (*Figure 1A*).

To perform structural studies, we mixed lysates from HEK293 cells that independently expressed human CD69 and human S1PR1. The CD69-S1PR1 complex was then incubated with $G_{\alpha i \beta 1 \gamma 2}$ heterotrimer and scFv16 (*Maeda et al., 2018*) at 1:1.2:1.4 molar ratio. After gel-filtration purification, the resulting complex was concentrated for cryo-EM analysis (*Figure 1—figure supplement 1A*). We obtained over 1 million particles from ~4000 cryo-EM images. The overall structure of the CD69-bound S1PR1 coupled to heterotrimeric $G_i$ was determined at 3.15 Å resolution by 293,516 particles (*Figure 1—figure supplement 1B–F*; *Table 1*). The structure shows that one S1PR1 binds one CD69 homodimer and one $G_i$ heterotrimer. It also revealed well-defined features for the canonical seven transmembrane helices (7-TMs) of S1PR1, the $G_{\alpha i}$ Ras-like domain, the $G_\beta$ and $G_\gamma$ subunits and scFv16 (*Figure 1B*, *Figure 1—figure supplement 2A*). The intracellular loop 3 (ICL3) and the C-terminus of S1PR1 and the intracellular and extracellular domains of CD69 were not found in the cryo-EM map indicating their flexibility in the complex. In contrast, the TMs of the CD69 homodimer were clearly

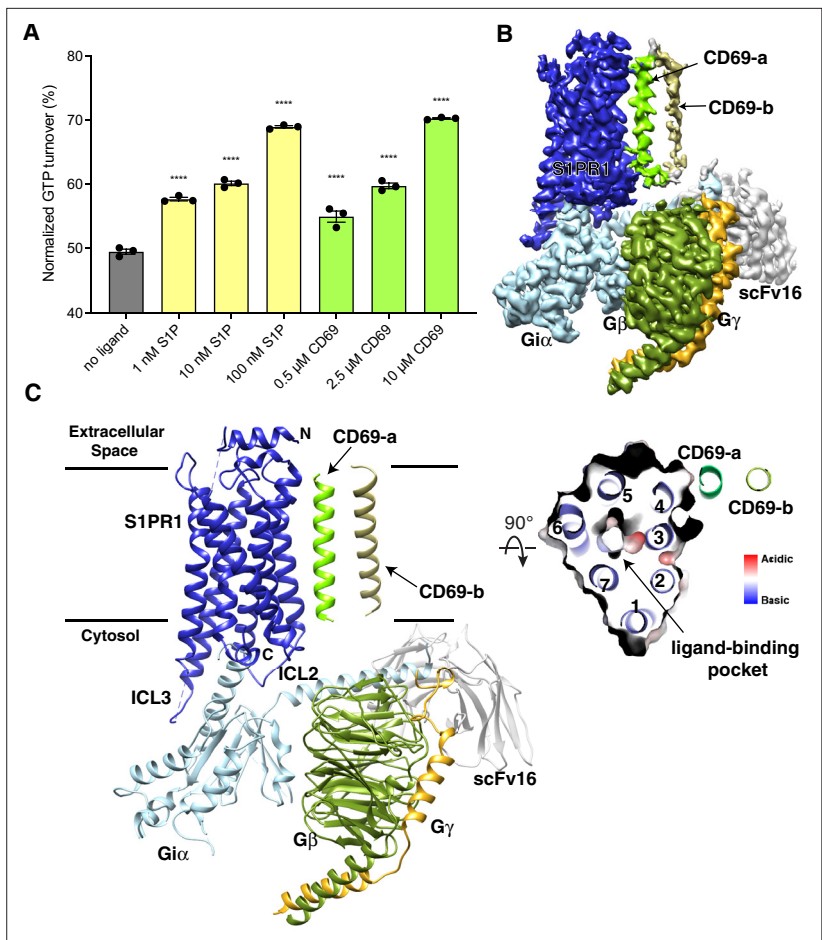

**Figure 1.** Overall structure of human CD69-S1PR1-$G_i$ complex. (**A**) S1PR1-induced GTP turnover for $G_{i1}$ in the presence of purified CD69 or S1P. Luminescence signals were normalized relative to the condition with $G_{i1}$ only. Data are mean ± s.e.m. of three independent experiments. One-way ANOVA with Tukey's test; ****$p<0.0001$. Experiments were repeated at least three times with similar results. (**B**) Cryo-EM map of human CD69 bound S1PR1-$G_i$ complex. (**C**) Cartoon presentation of the complex in the same view and color scheme as shown in (**B**). Slab view of S1PR1 from the extracellular side showing that the orthosteric binding pocket is vacant.

The online version of this article includes the following source data and figure supplement(s) for figure 1:

**Figure supplement 1.** Cryo-EM reconstruction of CD69 bound S1PR1-$G_i$ complex.

**Figure supplement 1—source data 1.** Original uncropped SDS-PAGE gels for data in *Figure 1—figure supplement 1*.

**Figure supplement 1—source data 2.** Uncropped SDS-PAGE gels for data in *Figure 1—figure supplement 1* with the relevant bands labeled.

**Figure supplement 2.** The cryo-EM density map of CD69-bound S1PR1-$G_i$ complex.

**Figure supplement 3.** Structural comparison between CD69-bound S1PR1 and S1P-bound S1PR1.

**Figure supplement 4.** Structures of homodimeric and heterodimeric GPCRs.

---

defined in the map owing to their interactions with S1PR1 (*Figure 1B*, *Figure 1—figure supplement 2A*; the interacting TM helix is referred to as CD69-a). Because no lipid was supplemented into the protein during the expression and purification, there is no notable lipid ligand in the 7-TM bundle of S1PR1, which is different from the previous structural discoveries on S1PRs (*Chen et al., 2022*; *Liu et al., 2022*; *Xu et al., 2022*; *Yu et al., 2022*; *Yuan et al., 2021*; *Zhao et al., 2022*).

Structural comparison shows that the entire complex and the S1P bound S1PR1-$G_i$ complex share a similar conformation with a root-mean-square deviation (RMSD) of 0.82 Å (*Figure 1—figure supplement 3A*). The receptors in both complexes can be aligned well; however, the F161$^{4.43}$ in TM4 presents

**Table 1.** Cryo-EM data collection, processing, and refinement statistics.

| Structure | CD69-S1PR1-Gi-scFv16 |
| --- | --- |
| PDB | 8G94 |
| EMDB | EMD-29861 |
| Data collection/ processing | |
| Magnification | 105,000 |
| Voltage (kV) | 300 |
| Pixel size (Å) | 0.83 |
| Defocus range (μm) | 1.0–2.0 |
| Electron exposure (e⁻/Å²) | 60 |
| Symmetry imposed | C1 |
| Initial particles (No.) | ~1.1 million |
| Final particles (No.) | 293,516 |
| Map resolution (Å) | 3.14 |
| FSC threshold | 0.143 |
| Map resolution range (Å) | 25–3.0 |
| Refinement | |
| Model Resolution (Å) | 3.3 |
| FSC threshold | 0.5 |
| Map sharpening B-factor (Å²) | –60 |
| Model composition | |
| Non-hydrogen atoms | 9223 |
| Protein residues | 1187 |
| Ligand | 0 |
| *B*-factors (Å²) | |
| Protein | 98.23 |
| R.m.s. deviations | |
| Bond lengths (Å) | 0.006 |
| Bond angles (°) | 0.702 |
| Validation | |
| MolProbity score | 1.64 |
| Clashscore | 6.49 |
| Rotamers outliers (%) | 0.00 |
| Ramachandran plot (%) | |
| Favored | 95.87 |
| Allowed | 4.13 |
| Outliers | 0.00 |

a notable shift due to CD69 binding (*Figure 1—figure supplement 3B*). We docked the S1PR1 bound to the other TM of the CD69 homodimer which showed that the modeled receptor would sterically clash with the $G_{i\alpha}$ subunit (*Figure 1—figure supplement 3C*). This may explain why only one receptor binds one CD69 homodimer in the presence of the heterotrimeric G-protein.

Receptor activity-modifying protein 1 (RAMP1), a type I transmembrane domain protein, binds the calcitonin receptor-like receptor (CLR) class B GPCR to form the Calcitonin gene-related peptide (CGRP) receptor which is involved in the pathology of migraine (*Russell et al., 2014*). The structure of $G_s$-protein coupled CGRP receptor uncovers that TM of RAMP1 interacts with TMs 3–5 of CLR and the extracellular domains of RAMP1 and CLR have extensive interactions (*Liang et al., 2018*; *Figure 1—figure supplement 4A*). Both CLR and RAMP1 contribute to the engagement of their agonist CGRP. However, in our structure, CD69 acts as an agonist to activate S1PR1 through a direct binding to TM4 of S1PR1 in the absence of a canonical agonist (e.g. S1P or FTY720-P). The extracellular domain of CD69 is completely invisible in the complex and may not interact with the extracellular loops of S1PR1.

Another type of intramembrane interaction observed for GPCRs is the formation of either homodimers or heterodimers. The metabotropic glutamate receptor 2 (mGlu2), a Class-C GPCR, employs TM4 to maintain its inactive dimeric state or TM6 to assemble as a homodimer in the presence of its agonist (*Du et al., 2021*; *Figure 1—figure supplement 4B*). The structure of inactive mGlu2–mGlu7 heterodimer shows that TM5 plays a key role in the complex assembly (*Du et al., 2021*; *Figure 1—figure supplement 4C*). Moreover, TM1 of the class D GPCR Ste2 is responsible for engaging the TM1 of another Ste2 to form a homodimer (*Velazhahan et al., 2021*; *Figure 1—figure supplement 4D*). These findings elucidate that GPCRs could employ distinct TMs to recruit their transmembrane binding partners.

The TM of one protomer of CD69 homodimer interacts with the TM4 of S1PR1 (*Figure 1C*). The interface area between TMs is about 600 Å². Structural analysis shows that residues V41, V45, V48, V49, T52, I56, I59, A60 of CD69 mediate its extensive interactions with the receptor (*Figure 2A*, *Figure 1—figure supplement 2B*). Residues L160[4.42], F161[4.43], I164[4.46], W168[4.50], V169[4.51], L172[4.54], I173[4.55], G176[4.58], I179[4.61] and M180[4.62] of S1PR1-TM4 contribute to the interaction with CD69 (*Figure 2B*, *Figure 1—figure supplement*

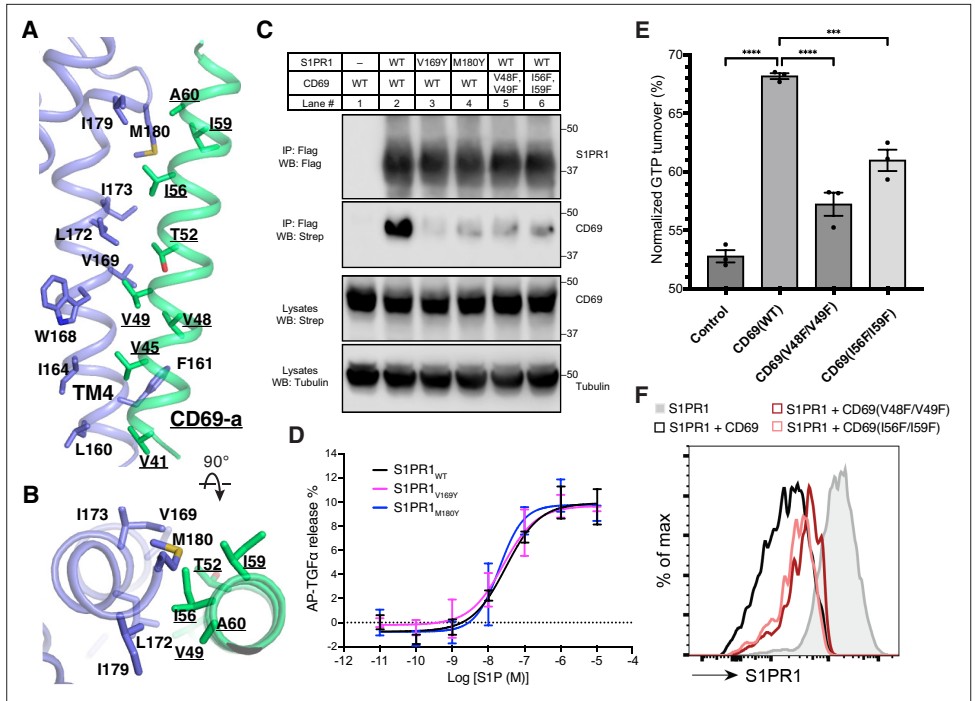

**Figure 2.** The binding interface between CD69 and S1PR1. (**A**) and (**B**) Detailed interactions between CD69-a and TM4 of S1PR1. Residues that contribute to complex formation are labeled. CD69 is shown in green and S1PR1 in slate. (**C**) S1PR1-Flag and CD69-StrepII co-immunoprecipitation assay in transfected HEK293 GnTI⁻ cells from one experiment that is representative of three. (**D**) Dose-response curves of S1PR1$_{WT}$, S1PR1$_{V169Y}$ and S1PR1$_{M180Y}$ for the TGFα shedding assay using S1P. Data are mean ± s.d. (n=3). (**E**) S1PR1-induced GTP turnover for G$_{i1}$ in the presence of purified wild-type and mutant CD69. Luminescence signals were normalized relative to the condition with G$_{i1}$ only. Data are mean ± s.e.m. of three independent experiments. One-way ANOVA with Tukey's test; ***p<0.001, ****p<0.0001. Experiments in (**C**)-(**E**) were repeated at least twice with similar results. (**F**) Flow cytometric analysis of S1PR1 surface expression on WEHI231 lymphoma cells transduced with S1PR1 and CD69 wild-type and mutant constructs as indicated. From one experiment that is representative of three.

The online version of this article includes the following source data and figure supplement(s) for figure 2:

**Source data 1.** Original uncropped western blots for data in *Figure 2*.

**Source data 2.** Uncropped western blots for data in *Figure 2* with the relevant bands labeled.

**Figure supplement 1.** Size exclusion column profiles of CD69 wild type and mutants.

**Figure supplement 1—source data 1.** Original uncropped SDS-PAGE gels for data in *Figure 2—figure supplement 1*.

**Figure supplement 1—source data 2.** Uncropped SDS-PAGE gels for data in *Figure 2—figure supplement 1* with the relevant bands labeled.

---

*2B*). However, the TM of another CD69 does not have any interactions with the receptor and heterotrimeric G$_i$ protein (*Figure 1C*). Further structural comparison with the S1P-bound S1PR1-G$_i$ complex indicates that the heterotrimeric G$_i$ proteins in both complexes exhibit a similar state with a RMSD of 0.45 Å. Also, the intracellular regions of the heptahelical domain adopt a similar conformation to accommodate the G$_i$ proteins. This finding implies that S1P and CD69 stimulate the receptor to engage the heterotrimeric G$_i$ proteins in an analogous fashion.

To validate our structural observations, we performed the co-immunoprecipitation (co-IP) assay using S1PR1 and CD69 variants. Compared to the wild-type S1PR1, two mutants (V169[4.51]Y and M180[4.62]Y) present reduced binding to CD69 (*Figure 2C*). The TGFα shedding assay showed that these two mutants retained normal activity in response to S1P (*Figure 2D*). We also tested two CD69 double mutations (V48F/V49F and I56F/I59F) for their association with S1PR1. The co-IP results show that the interaction between S1PR1 and either mutant is considerably attenuated, thus directly supporting the role of CD69-TM in the complex assembly (*Figure 2C*). Moreover, we have purified

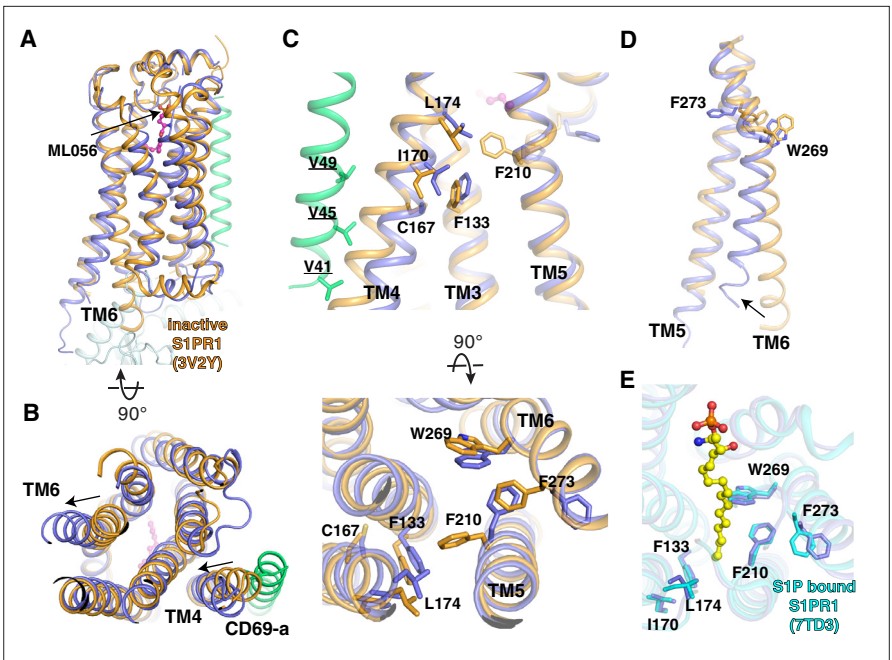

**Figure 3.** Comparison between CD69-bound S1PR1 and ML056- or S1P-bound S1PR1. (**A**) Overall structures of S1PR1 binding with CD69 and ML056. The CD69-bound S1PR1 structure was aligned to ML056-bound inactive S1PR1 (PDB code: 3V2Y). ML056-bound receptor is shown in brown, CD69-bound receptor in blue, and the TM of CD69 in green. The same color scheme is used (**C**) and (**D**). (**B**) The movements of TM4 and TM6 of CD69-bound S1PR1 compared with ML056-bound inactive S1PR1. (**C**) Residues involved in the TM movements. (**D**) TM6 movement around W269$^{6.48}$ and F273$^{6.52}$. (**E**) Comparison between CD69-bound S1PR1 and S1P-bound S1PR1 (PDB code: 7TD3). Residues in the ligand binding pocket are shown. CD69-bound receptor in blue and S1P-bound S1PR1 in cyan. S1P is shown as balls and sticks in yellow.

The online version of this article includes the following source data and figure supplement(s) for figure 3:

**Figure supplement 1.** S1PR1 specificity for CD69 binding.

**Figure supplement 1—source data 1.** Original uncropped western blots for data in *Figure 3—figure supplement 1*.

**Figure supplement 1—source data 2.** Uncropped western blots for data in *Figure 3—figure supplement 1* with the relevant bands labeled.

**Figure supplement 2.** Structures of GPCRs with their positive allosteric modulators.

CD69(V48F/V49F) and CD69(I56F/I59F) individually and mixed with S1PR1 and G$_{\alpha i\beta 1\gamma 2}$ to conduct a GTPase-Glo assay (*Figure 2—figure supplement 1*). Consistent with the results of our co-IP assays, the activation of G$_i$ proteins in the presence of either variant was decreased (*Figure 2E*). To further validate the physiological role of the CD69-S1PR1-G$_i$ complex, we tested the two CD69 variants, for their influence on CD69-mediated S1PR1 internalization in WEHI231 B lymphoma cells. In accord with the biochemical data, CD69(V48F/V49F), and CD69(I56F/I59F) were both reduced in their ability to downregulate S1PR1 (*Figure 2F*).

The structures of the S1PR1 complex with its small molecule modulators (including S1P, FTY720-P, BAF312, and ML056) uncover that the TMs 3, 5, 6, and 7 contribute to accommodate the modulators in the orthosteric site (*Hanson et al., 2012*; *Liu et al., 2022*; *Xu et al., 2022*; *Yu et al., 2022*; *Yuan et al., 2021*). In contrast, S1PR1 employs its TM4 to associate with CD69 which functions as a protein agonist for triggering receptor activation. Structural comparison with the inactive state of ML056 bound S1PR1 reveals a unique mechanism of CD69-mediated S1PR1 activation (*Figure 3A*). The binding of CD69 induces a 4 Å shift at the intracellular end of TM4 causing the residues C167$^{4.49}$, I170$^{4.52}$, and L174$^{4.56}$ in TM4 to face TM3 (*Figure 3B*). C167$^{4.49}$ and I170$^{4.52}$ have hydrophobic contacts with the F133$^{3.41}$ in TM3, and L174$^{4.56}$ pushes the F210$^{5.47}$ in TM5 towards the edge of the receptor to form the hydrophobic interactions with W269$^{6.48}$ and F273$^{6.52}$ in TM6 (*Figure 3C*, *Figure 1—figure supplement 2C*). These interactions trigger the notable

movement of TM5 and TM6 allowing the opening of intracellular regions to engage the hetero-trimeric $G_i$ proteins (*Figure 3A and D*). Residues A137[3.45], I170[4.52], L174[4.56], F210[5.47], W269[6.48], and F273[6.52] are conserved among S1PR1, S1PR2 and S1PR3, but not F133[3.41] and C167[4.49]. Remarkably, further comparison shows that the key residues, which are crucial for the S1P binding and receptor activation, present similar conformations in the structures of S1P-bound S1PR1 and CD69-bound S1PR1, although S1P and CD69 have different structural natures and completely distinct binding sites in the receptor (*Figure 3E*).

To date, five S1PRs have been identified. These receptors have different tissue distributions, and they also function via distinct kinds of G proteins (including $G_i$, $G_q$, and $G_{12/13}$) (*Cartier and Hla, 2019*). Previous work showed that CD69 specifically binds to S1PR1, and it does not associate with S1PR2, S1PR3 or S1PR5 (*Bankovich et al., 2010*; *Jenne et al., 2009*; *Shiow et al., 2006*). To dissect the binding specificity of CD69, we carried out the co-IP assays to show a very weak interaction between S1PR2 and CD69 (*Figure 3—figure supplement 1A*). Although the sequence homology among five S1PRs is high, residues L157[4.51] and L168[4.62] in S1PR2-TM4 are not conserved with those in S1PR1 and are determinants for specific recognition of CD69 (*Figure 3—figure supplement 1B*). We speculated that converting these two residues to those in S1PR1 may prompt the interaction between S1PR2 variant and CD69. Our co-IP result clearly shows that S1PR2(L157[4.51]V/L168[4.62]M) could interact with CD69 albeit the interactions are weaker than that between S1PR1 and CD69 (*Figure 3—figure supplement 1A*). This finding further demonstrates the essential role of S1PR1-TM4 in the CD69-mediated S1PR1 signaling.

All the known small molecule S1PR1 agonists or antagonists bind to the orthosteric site in the heptahelical domain (*Hanson et al., 2012*; *Liu et al., 2022*; *Xu et al., 2022*; *Yu et al., 2022*; *Yuan et al., 2021*). Interestingly, the CD69 binding site is akin to that of the allosteric agents which attach to receptors (*Draper-Joyce et al., 2021*; *Mao et al., 2020*; *Yang et al., 2022*), although the nature of these agents and CD69 is quite different. The diversity of the allosteric modulator binding sites in GPCRs has been revealed by numerous structures (*Figure 3—figure supplement 2*). When the orthosteric site is occupied, the positive allosteric modulator attaches to the receptor and then increases agonist affinity and/or efficacy. CD69 binds to the edge of S1PR1, but it acts as a protein agonist to directly activate the receptor in the absence of any agonists in the orthosteric site. Thus, our finding suggests CD69 is different from other S1PR1 agonists in that it functions via a direct binding to the edge of the receptor.

It remains unknown whether the antagonist of S1PR1 bound to the 7-TMs will affect the CD69-mediated regulation of S1PR1. We co-transfected S1PR1-GFP and CD69-mCherry into HEK293 cells in a lipid depleted medium. After 24 hr, the fluorescence images show that substantial receptors (~80%) have been internalized with CD69. However, when we added the Ex26, a potent S1PR1 antagonist (*Cahalan et al., 2013*), into the cells 6 hr after transfection, the images show that just ~50% receptors have been internalized (*Figure 4A and B*). This finding indicates that the CD69-mediated S1PR1 activation could be reversed when the 7-TMs pocket is preoccupied by an antagonist.

It has been known that S1P, FTY720-P and CD69, could promote the internalization of S1PR1. However, the mechanisms of S1P- and FTY720-P-mediated internalization appear to be different. While both S1P and FTY720-P activate $G_i$-signaling, FTY720-P is considered as a β-arrestin-biased agonist, and FTY720-P-induced S1PR1 internalization is β-arrestin-dependent (*Oo et al., 2007*; *Xu et al., 2022*). The pathway of S1P-mediated internalization can be β-arrestin-dependent or independent (*Galvani et al., 2015*; *Reeves et al., 2016*). To test the mechanism of how CD69 induces the receptor internalization, we performed a fluorescence imaging assay to check the internalization of S1PR1 in the presence of either $G_i$ inhibitor Pertussis toxin (PTX) or β-arrestin inhibitor Barbadin. The plasmids encoding S1PR1-GFP and CD69-mCherry were co-transfected into HEK293 cells in a lipid depleted medium. After 6 hr, we added PTX or Barbadin. On day 2, we calculated the fraction of internalized S1PR1 in each group by fluorescence imaging. The results show that Barbadin does not interfere with CD69-induced receptor internalization (*Figure 4C and D*), but PTX could prevent half of the receptors from internalization (*Figure 4E and F*). Our finding also supports that Barbadin was effective in reducing FTY70-P-induced S1PR1 internalization (*Figure 4—figure supplement 1*). Thus, CD69 agonism of S1PR1 induces $G_i$-dependent internalization of the complex.

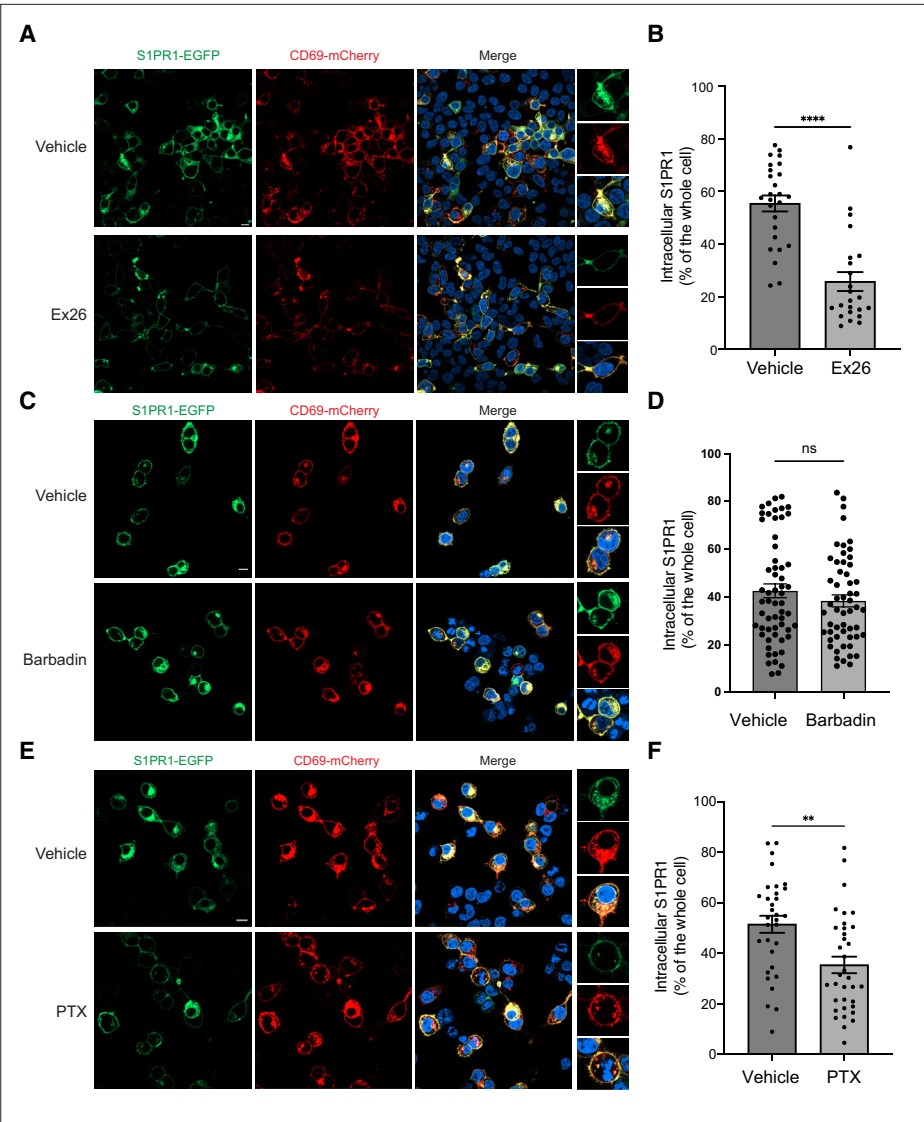

**Figure 4.** CD69 induced S1PR1 internalization. (**A**) HEK293 cells were treated with 2 μM Ex26 or vehicle for 12 hr and imaged using confocal microscopy. Scale bar, 10 μm. (**B**) Quantification of intracellular S1PR1 of the cells in (**A**). (**C**) HEK293 cells were treated with 20 μM Barbadin for 12 hr and imaged for analysis. Scale bar, 10 μm. (**D**) Quantification of intracellular S1PR1 of the cells in (**C**). (**E**) HEK293 cells were treated with 200 ng/ml pertussis toxin (PTX) for 12 hr and imaged for analysis. Scale bar, 10 μm. (**F**) Quantification of intracellular S1PR1 of the cells in (**E**). Data are mean ± s.e.m. Two-sided Welch's t-test; ns, not significant, **p<0.01, ****p<0.0001. All experiments were repeated at least three times with similar results.

The online version of this article includes the following figure supplement(s) for figure 4:

**Figure supplement 1.** Barbadin alters the FTY720-P mediated S1PR1 internalization.

**Figure supplement 2.** S1PR1-induced GTP turnover for $G_{i1}$ in the presence of purified CD69 and S1P.

## Discussion

Our studies provide a model for understanding how the lymphocyte activation marker CD69 controls lymphocyte egress and thus augments adaptive immunity. As an immediate early gene, CD69 is strongly transcriptionally induced in lymphocytes within an hour of exposure to type I IFN, toll-like receptor (TLR) ligands, or antigen receptor engagement (*Grigorova et al., 2010*; *Shiow et al., 2006*; *Ziegler et al., 1994*). Following induction, CD69 protein engages S1PR1 as an agonist, causing S1PR1 internalization and loss of the ability to sense S1P gradients. We speculate that even prior to internalization, CD69 disrupts S1PR1's egress promoting function by acting as a high concentration agonist

and thus making the receptor 'blind' to S1P distribution. Consistently, our functional analysis reveals that CD69 could not synergize with S1P to trigger S1PR1 activation (*Figure 4—figure supplement 2*).

Previous work has shown the critical importance of correctly distributed S1P and thus correctly localized S1PR1 activation for effective lymphocyte egress (*Schwab et al., 2005*). As well as promoting egress, S1PR1, transmits signals needed for maintaining T cell survival (*Mendoza et al., 2017*) and CD69 has been implicated in transmitting signals that influence T cell differentiation (*Cibrián and Sánchez-Madrid, 2017*; *Kimura et al., 2017*). Whether the CD69-S1PR1 complex contributes to these signals before undergoing degradation merits further study. GRK2 (*Arnon et al., 2011*; *Oo et al., 2007*; *Watterson et al., 2002*) and dynamin (*Willinger et al., 2014*) participate in S1PR1 internalization in response to S1P. In accord with these factors possibly having a role in CD69-mediated S1PR1 internalization, they have been shown to promote internalization of some receptors independently of β-arrestins (*Moo et al., 2021*). The selectivity of CD69 for S1PR1 is important for allowing activated CD69[+] lymphocytes and natural killer cells to employ other S1PRs, such as S1PR2 and S1PR5, to carry out functions without interruption by CD69 (*Jenne et al., 2009*; *Laidlaw et al., 2019*; *Moriyama et al., 2014*). The lack of conservation of key residues that mediate the S1PR1-CD69 interaction in TM4 of S1PR2, S1PR5 and the other S1PRs provides an explanation for this selectivity (*Figure 3— figure supplement 1B*). In summary, we provide the first example of GPCR activation by interaction in cis with a transmembrane ligand and thereby explain the mechanism of lymphocyte egress shutdown. The structure also offers insights that may enable introduction of transcriptionally inducible GPCR switches into CAR-T cells and other engineered cell types.

## Methods
### Constructs
For expression and purification, the wild-type human S1PR1 (a.a.1–347, UniProt: P21453) and CD69 (full-length, UniProt: Q07108) were separately cloned into pEZT-BM vector (*Morales-Perez et al., 2016*) with a C-terminal Flag tag and StrepII tag, respectively. Plasmids of Gα$_{i1}$, Gβ$_1$/Gγ$_2$ and scFv16 are kind gifts from Brian Kobilka (Stanford University). For co-immunoprecipitation assay, the full-length wild-type human S1PR1 fused with a C-terminal Flag tag and CD69 fused with a C-terminal StrepII tag, were separately cloned into pCAGGS vector (*Niwa et al., 1991*) with modified multiple cloning sites. For fluorescence microscopy, the plasmids pCAGGS-S1PR1-Flag-GFP and pCAGGS-CD69-StrepII-mCherry were constructed.

### Protein expression and purification
S1PR1-Flag and CD69-StrepII were separately expressed using baculovirus-mediated transduction of mammalian HEK293S GnTI⁻ cells (ATCC CRL-3022) in a medium containing FreeStyle 293 (Gibco Cat# 12338018) supplemented with 2% charcoal-dextran stripped fetal bovine serum (Gibco Cat# 12676029), penicillin (100 U/mL), and streptomycin (100 µg/mL) (Corning Cat# 30–002 CI). Baculoviruses were generated in Sf9 cells, and P2 virus was used to infect HEK293S GnTI⁻ cells at 37 °C. At 8 hr after infection, sodium butyrate at a final concentration of 10 mM was added to the culture. After further incubation for 64 hr at 30 °C, cells expressing S1PR1-Flag and CD69-StrepII were mixed together and resuspended in buffer A (20 mM HEPES, pH 7.5, 150 mM NaCl) supplemented with protease inhibitors and then homogenized by sonication. The protein was solubilized with 1% LMNG (lauryl maltose neopentyl glycol) /0.1% CHS (cholesteryl hemisuccinate) for 1 hr at 4 °C. Insoluble material was removed by centrifugation at 40,000 *g*, 4 °C for 30 min, and the supernatant was incubated with Strep-Tactin XT resin (IBA Cat# 2-5030-025) for batch binding. The resin was washed with 20 column volumes (CV) of buffer A containing 0.01% LMNG/0.001% CHS. The protein complex was eluted with 6 CVs of buffer A containing 0.01% LMNG/0.001% CHS and 50 mM biotin, followed by a second affinity purification by anti-Flag M2 resin (Sigma-Aldrich). The excessive CD69-StrepII was washed off with 20 CVs of buffer A containing 0.01% LMNG/0.001% CHS, and the complex was eluted with 5 CVs of 3×Flag peptide (0.1 mg/ml; ApexBio). The eluted protein was further purified by gel filtration using a Superose 6 Increase 10/300 GL column (Cytiva) with 20 mM HEPES, pH 7.5, 150 mM NaCl, 0.001% L-MNG/0.0001% CHS, and 0.0025% glyco-diosgenin (GDN). The peak fractions were collected for complex assembly.

To assemble the CD69-S1PR1-G$_i$-scFv16 complex, purified CD69-S1PR1 was mixed with the Gi heterotrimer at a 1:1.2 molar ratio. This mixture was incubated on ice for 1 hr, followed by the addition of apyrase to catalyze the hydrolysis of unbound GDP on ice for 1 hr. Then, scFv16 was added at a 1.4:1 molar ratio (scFv16: CD69-S1PR1) followed by 30 min incubation on ice. The mixture was diluted 10-fold by gel filtration column buffer. To remove excess Gi and scFv16 proteins, the mixture was purified by anti-Flag M2 affinity chromatography. The complex was eluted and concentrated using an Amicon Ultra Centrifugal Filter (molecular weight cutoff 100 kDa). The complex was further purified by gel filtration (Superose 6 Increase 10/300 GL) with buffer 20 mM HEPES, pH 7.5, 150 mM NaCl, 0.001% L-MNG/0.0001% CHS, and 0.0025% GDN. Peak fractions consisting of CD69-S1PR1-G$_i$ complex were concentrated to ~10–12 mg/ml for cryo-EM studies.

## Cryo-EM sample preparation and data acquisition

The freshly purified CD69-S1PR1-G$_i$-scFv16 complex was added to Quantifoil R1.2/1.3 400-mesh Au holey carbon grids (Quantifoil), blotted using a Vitrobot Mark IV (FEI), and vitrified in liquid ethane. The grids were imaged in a 300-kV Titan Krios (FEI) with a Gatan K3 Summit direct electron detector. Data were collected in super-resolution mode at a pixel size of 0.415 Å with a dose rate of 23 electrons per physical pixel per second. Images were recorded for 5 s exposures in 50 subframes with a total dose of 60 electrons per Å$^2$.

## Imaging processing and 3D reconstruction

A total of 4,239 dose-fractionated image stacks of CD69-S1PR1-G$_i$ complex were collected and subjected to single particle analysis using RELION-3.1 (*Zivanov et al., 2018*) and cryoSPARC v3.3 (*Punjani et al., 2017*). MotionCor2 (*Zheng et al., 2017*) was used for motion correction and dose weighting, CTFFIND-4.1 *Rohou and Grigorieff, 2015* for contrast transfer function (CTF) estimation, and crYOLO *Wagner et al., 2019* for particle picking with a general model. A total of 1,113,446 particles were extracted with a pixel size of 1.66 Å in RELION and imported to cryoSPARC. The imported particles were subjected to ab initio model reconstruction and several rounds of alternating 2D classification and heterogeneous refinement. Then 336,669 particles from the best class were re-extracted at full pixel size (0.83 Å) in RELION and imported to cryoSPARC again. Two heterogeneous refinements were performed in parallel and the resulting particles from the two best classes were combined with duplicates removed. These 293,516 particles were subjected to CTF refinement and Bayesian polishing followed by masked 3D auto refinement. RELION postprocessing was used for sharpening of the final map.

## Model construction and refinement

The cryo-EM structure of the S1PR1-G$_i$ bound to S1P (PDB: 7TD3) (*Liu et al., 2022*) was used as initial models and manually docked into cryo-EM density map with UCSF Chimera-1.15 (*Pettersen et al., 2004*). The transmembrane helix of CD69 was manually built using Coot-0.9.6 (*Emsley and Cowtan, 2004*). Due to the limited local resolution, the TM of CD69-b was built as polyalanine. The resulting model was subjected to iterative rounds of manual adjustment and rebuilding in Coot and real-space refinement in Phenix-1.16 (*Adams et al., 2010*). MolProbity (*Williams et al., 2018*) was used to validate the geometries of the model. Structural figures were generated using UCSF Chimera-1.15, ChimeraX-1.5 (*Pettersen et al., 2021*), and PyMOL-2.3 (https://pymol.org/2/).

## GTP turnover assay

GTP turnover was analyzed using GTPase-Glo Assay kit (Promega Cat# V7681). Briefly, the purified S1PR1 was first incubated with purified CD69 and/or S1P followed by mixing with isolated Gi protein in an assay buffer containing 20 mM HEPES, pH7.5, 150 mM NaCl, 0.01% LMNG/0.001% CHS, 10 mM MgCl$_2$, 100 µM TCEP, 10 µM GDP and 5 µM GTP. After incubation for 60 min, the reconstituted GTPase-Glo reagent was added to the sample and incubated for 30 min at room temperature. The amount of remaining GTP was assessed by measuring luminescence after adding and incubation with the detection reagent for 10 min at room temperature. The luminescence signal was normalized in each case to that of G-protein alone. Data were analyzed using GraphPad Prism 9.

## Co-immunoprecipitation and immunoblotting assay

HEK293 GnTI$^-$ cells were transfected with plasmids encoding CD69-StrepII and S1PR1-Flag using FuGene 6 transfection reagent in 60 mm dishes. Forty-eight hr post transfection, cells were harvested

and whole cell lysates were prepared using IP lysis buffer (Thermo Scientific) supplemented with protease inhibitor cocktail (Roche). Lysates were cleared by centrifuging at 20,000 $g$ for 15 min at 4 °C. Supernatants were incubated with anti-Flag M2 affinity beads (MilliporeSigma) with end-over-end rotation for 2 h at 4 °C. Beads were washed three times with lysis buffer for 5 min per wash with end-over-end rotation at 4 °C. Proteins were eluted from beads with lysis buffer supplemented with 0.4 mg/ml 3×Flag peptide. Protein samples were loaded Bolt 4–12% Bis-Tris plus gels (Invitrogen) and transferred to TransBlot Turbo nitrocellulose membranes (Bio-Rad). Membranes were blocked for 1 hr at room temperature with 5% milk in PBS with 0.05% Tween 20 (PBST) followed by primary antibody incubation, three-times wash, secondary antibody incubation, and three-times wash again. Membranes were developed for 2 min at room temperature using SuperSignal West Pico PLUS Chemiluminescent Substrate (Thermo Scientific) and then imaged using the LI-COR Odyssey Fc imaging system. The following primary antibodies were used: Tubulin (D3U1W), Cell Signaling Cat# 86298 (1:3000 dilution); Flag tag (FLA-1), MBL International Cat# M185-3L (1:3000 dilution); StrepII tag, IBA GmbH Cat# 2-1507-001 (1:2000 dilution). Anti-mouse IgG HRP-linked secondary antibody (Cell Signaling Cat# 7076) was used for chemiluminescent detection (1:3000 dilution).

## TGFα shedding assay

The agonist activity of S1P for the mutant S1PR1s was determined by the TGFα shedding assays (*Inoue et al., 2012*). Briefly, three pCAGGS plasmids encoding the human full-length S1PR1 variant (empty vector as negative control), the chimeric $G_{\alpha q/i1}$ subunit and alkaline phosphatase-fused TGFα (AP-TGFα) were co-transfected into HEK293 cells using FuGene 6 transfection reagent in a 12-well plate. After 24 hr, the transfected cells were collected by trypsinization, washed with phosphate-buffered saline (PBS), and resuspended in Hanks' balanced salt solution (HBSS) with 5 mM HEPES (pH 7.4). Then, the cells were seeded into a 96-well culture plate and treated with S1P, which was serially diluted in HEPES-containing HBSS with 0.01% fatty acid–free bovine serum albumin. After incubation with S1P, the cell plate was spun, and conditioned media was transferred to an empty 96-well plate. AP reaction solution (120 mM Tris-HCl, pH 9.5, 40 mM NaCl, 10 mM $MgCl_2$, and 10 mM p-nitrophenyl phosphate) was added into the cell plates and the conditioned media plates. The absorbance at 405 nm was measured using a microplate reader (Synergy Neo2, BioTek) before and after 2 hr incubation at 37 °C. Ligand-induced AP-TGFα release was calculated as described previously (*Inoue et al., 2012*). AP-TGFα release signal of empty vector-transfected cells were subtracted from that of S1PR1 cells at the corresponding S1P concentration points. Then, the vehicle-treated AP-TGFα release signal was set as a baseline and ligand-induced AP-TGFα release signals were fitted to a four-parameter sigmoidal concentration–response curve using GraphPad Prism 9 software.

## Fluorescence microscopy

HEK293 cells were plated in 35 mm glass bottom dishes (Cellvis Cat# D35141.5N) followed by transfection with S1PR1-GFP and/or CD69-mCherry using FuGene 6 reagent on the next day. Twenty-four hr post transfection, the cells were stained with Hoechst 33342 reagent (Thermo Fisher Cat# R37605) and fluorescence images were acquired using a Zeiss LSM 800 microscope system with ZEN imaging software (Zeiss).

For fluorescence quantification of intracellular S1PR1 and CD69, outside and inside of plasma membrane were circled manually in Fiji software (*Schindelin et al., 2012*). The fluorescence intensities in each circle were measured and regarded as whole-cell and intracellular fluorescence intensity, respectively. The intracellular fluorescence intensity was normalized to its corresponding whole-cell fluorescence intensity. For each data point, ~30 cells were randomly selected for quantification. The data shown in the figures are representative of two or more independent experiments.

## WEHI231 cell retroviral transduction

WEHI231 B lymphoma cells were co-transduced with retroviral constructs encoding OX56 N-terminal tagged human S1PR1 containing an IRES-hCD4 reporter and either empty vector or constructs encoding wildtype, V48F/V49F or I56F/I59F human CD69 and an IRES-GFP reporter using methods previously described (*Lu et al., 2019*). After 3–5 days, the cells were harvested and rested for 20 min at 37 °C in PBS without serum, then stained to detect OX56, CD69, and hCD4. OX56 (S1PR1) staining on hCD4 +GFP + CD69 + cells were plotted.

## Acknowledgements

The data were collected at the UT Southwestern Medical Center Cryo-EM Facility (funded in part by the CPRIT Core Facility Support Award RP170644). We thank L Beatty, L Esparza, and Y Xu for technical support. This work was supported by NIH P01 HL160487, R01 GM135343, and Welch Foundation (I-1957) (to XL) and R01 AI040098 (to JGC). JGC is an investigator of Howard Hughes Medical Institute. This article is subject to HHMI's Open Access to Publications policy. HHMI lab heads have previously granted a nonexclusive CC BY 4.0 license to the public and a sublicensable license to HHMI in their research articles. Pursuant to those licenses, the author-accepted manuscript of this article can be made freely available under a CC BY 4.0 license immediately upon publication.

## Additional information

### Funding

| Funder | Grant reference number | Author |
| --- | --- | --- |
| National Institutes of Health | P01 HL160487 | Xiaochun Li |
| National Institutes of Health | R01 GM135343 | Xiaochun Li |
| Welch Foundation | I-1957 | Xiaochun Li |
| Howard Hughes Medical Institute | | Jason G Cyster |
| National Institutes of Health | R01 AI040098 | Jason G Cyster |

The funders had no role in study design, data collection and interpretation, or the decision to submit the work for publication.

### Author contributions

Hongwen Chen, Conceptualization, Investigation, Writing – review and editing; Yu Qin, Marissa Chou, Investigation, Writing – review and editing; Jason G Cyster, Xiaochun Li, Conceptualization, Supervision, Funding acquisition, Writing – original draft, Writing – review and editing

### Author ORCIDs

Hongwen Chen http://orcid.org/0000-0002-1065-9808
Jason G Cyster http://orcid.org/0000-0002-4735-9745
Xiaochun Li http://orcid.org/0000-0002-0177-0803

### Decision letter and Author response

Decision letter https://doi.org/10.7554/eLife.88204.sa1
Author response https://doi.org/10.7554/eLife.88204.sa2

## Additional files

### Supplementary files
• MDAR checklist

### Data availability

The 3D cryo-EM density maps have been deposited in the Electron Microscopy Data Bank under the accession number EMD-29861. Atomic coordinates for the atomic model have been deposited in the Protein Data Bank under the accession number 8G94. All other data needed to evaluate the conclusions in the paper are present in the paper and/or the supplementary materials.

The following datasets were generated:

| Author(s) | Year | Dataset title | Dataset URL | Database and Identifier |
|-----------|------|---------------|-------------|-------------------------|
| Chen H, Li X | 2023 | Structure of CD69-bound S1PR1 coupled to heterotrimeric Gi | https://www.rcsb.org/structure/8G94 | RCSB Protein Data Bank, 8G94 |
| Chen H, Li X | 2023 | Structure of CD69-bound S1PR1 coupled to heterotrimeric Gi | https://www.ebi.ac.uk/emdb/EMD-29861 | EMBD, EMD-29861 |

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
