## [Editor Report]

This important study provides unprecedented molecular insight into the activation and internalization of an important cell surface receptor induced by another membrane protein. The data supporting the conclusions are compelling, which include rigorous electron microscopy analysis, and biochemical and cell-based functional assays. The findings here not only reveal important mechanisms of S1P GPCR regulation, but also have implications for other fields such as receptor pharmacology and immunity.

---

## [Decision Letter]

**Decision letter after peer review:**

Thank you for submitting your article "Transmembrane protein CD69 acts as an S1PR1 agonist" for consideration by *eLife*. Your article has been reviewed by 2 peer reviewers, and the assessment has been overseen by a Reviewing Editor and Tadatsugu Taniguchi as the Senior Editor.

*Reviewing Editor (Recommendations for the Authors):*

1. Please show densities for the key residues of the binding interface between CD69 and S1PR1 (Figure 2a and b), and the key residues relay the movement from TM4 to TM6 upon CD69 binding (Figure 3c)

*Reviewer #1 (Recommendations for the Authors):*

Chen et al. solve the structure of S1PR1/CD69/Gi protein complex by Cryo-EM. They show that a single helix of CD69 contacts the TM4 domain of S1PR1, maps the interacting sites and demonstrates the activation of Gi protein by this "allosteric" transmembrane modulator of S1PR1 GPCR. Their work suggests that CD69 activates S1PR1 in a manner independent of the ligand S1P, which is surprising and perhaps somewhat different than a previous publication (PMID: 20463015). Mutagenesis studies define important residues in both S1PR1 TM4 and CD69 for the association. Through the use of inhibitors for Gi and ß-arrestin, they suggest that CD69 induces S1PR1 internalization via the Gi pathway. They also address why CD69 is selective for S1PR1 vs. other S1PRs (i.e. S1PR2-5). By mutating 2 residues in the TM4 of one of the non-interacting GPCR (S1PR2), the authors demonstrate a partial association, suggesting that this approach may be useful in modulating other GPCRs.

The major strength of this work is the provision of a high-resolution (3.5 Ä) structure of S1PR1/CD69/Gi complex. This is important since it is the first structural description of the transmembrane GPCR modulator. Given that S1PR1 is an essential regulator of vascular development and lymphocyte trafficking, and that S1PR1 modulators are an important class of drugs for autoimmune diseases, the implications of this work are quite widespread. The work is rigorous and definitive and the manuscript is written in a lucid and scholarly manner.

There are some minor issues regarding detailed conclusions that the authors might want to address, either by discussion and/or provision of new experimental data to further increase the impact of the work.

Suggestions for further discussion and/or provision of new data:

1) The non-requirement for S1P in the GTPase assay and the structure is interesting and novel, especially since a previous influential publication on CD69/ S1PR1 interaction (PMID: 20463015) suggested that not to be the case. This should be further expounded.

2) In Figure 1 – since there is some basal GTP turnover even without the ligand, I wonder if there is some trace S1P in the cell system, even though the cells are maintained in charcoal/dextran-stripped FBS. Do the authors know if the media is indeed S1P null? Similarly, cells can make S1P via SPT pathway and activate the receptor. Again, this could be trace amounts since the palmitate concentration in stripped FBS is likely low. It would strengthen the authors' argument to measure S1P in cells and media.

3) Figure 4- The internalization data could be strengthened if the high-resolution images of cells are provided as an inset.

4) Can CD69 dimer accommodate 2 S1PR1 molecules in the absence of G proteins? Similarly, can CD69/ S1PR1 complex (either monomer or dimer) accommodate ß-arrestin? I am not suggesting the provision of actual data, but based on the structures, modeling can suggest whether steric hindrance can permit such a complex.

*Reviewer #2 (Recommendations for the Authors):*

This paper by Chen et al. reports the functional characterization of a membrane protein, CD69, as an agonist of the S1p1 receptor to induce receptor activation and internalization without lipid ligands. The authors further determined a cryo-electron microscopy (cryo-EM) structure of the human S1p1-Gi complex with CD69. The structure revealed how CD69 activates S1p1 by interacting with the transmembrane helix 4 (TM4) of S1p1. The structural findings are further validated by mutagenesis studies. The results provide a novel mechanism for the modulation of GPCR signaling by other membrane proteins.

The findings and conclusions of this paper are well supported by data. The cryo-EM structure determination looks solid. All functional assays were performed with appropriate replicates and controls. I would suggest addressing or clarifying the following issues:

1) Does CD69 just function as an agonist of S1p1 in vivo? What are other possible functional roles of CD69 involving its extracellular region? Does CD69 need to form a homodimer to activate S1p1?

2) The Gi-induced and b-arrestin-independent receptor internalization is interesting. What could be the molecular mechanism? Gi protein itself doesn't internalize upon activation.

The data are solid. The writing is good. All figures are clear and easy to understand. I don't have further suggestions for the authors.

---

## [Author Response]

Reviewing Editor (Recommendations for the Authors):1. Please show densities for the key residues of the binding interface between CD69 and S1PR1 (Figure 2a and b), and the key residues relay the movement from TM4 to TM6 upon CD69 binding (Figure 3c)

The density has been shown in Figure 1—figure supplement 2.

Reviewer #1 (Recommendations for the Authors):There are some minor issues regarding detailed conclusions that the authors might want to address, either by discussion and/or provision of new experimental data to further increase the impact of the work.Suggestions for further discussion and/or provision of new data:1) The non-requirement for S1P in the GTPase assay and the structure is interesting and novel, especially since a previous influential publication on CD69/ S1PR1 interaction (PMID: 20463015) suggested that not to be the case. This should be further expounded.

The cited publication (by Bankovich, Shiow and Cyster) found that CD69-S1PR1 co-expression in a cell line led to enhanced S1PR1 binding of radiolabeled S1P and proposed ‘these findings fit best with the interpretation that CD69 stabilizes a high affinity ligand binding conformation of S1PR1’. Our current finding that S1PR1 in complex with CD69 mirrors the S1P bound state of S1PR1 is entirely consistent with the earlier observations. We also note that the earlier study provided biochemical evidence that TM4 of S1PR1 was involved in the interaction with CD69, in accord with our structural findings.

2) In Figure 1 – since there is some basal GTP turnover even without the ligand, I wonder if there is some trace S1P in the cell system, even though the cells are maintained in charcoal/dextran-stripped FBS. Do the authors know if the media is indeed S1P null? Similarly, cells can make S1P via SPT pathway and activate the receptor. Again, this could be trace amounts since the palmitate concentration in stripped FBS is likely low. It would strengthen the authors' argument to measure S1P in cells and media.

We agree that the presence of low amounts of S1P in the culture is difficult to exclude. However, we know that under these culture conditions, S1PR1 is expressed on the cell surface (when expressed without CD69) and is downmodulated by the addition of exogenous S1P. This finding suggests that the receptor cannot be fully occupied by any S1P that could be available in the cultures.

3) Figure 4- The internalization data could be strengthened if the high-resolution images of cells are provided as an inset.

Done.

4) Can CD69 dimer accommodate 2 S1PR1 molecules in the absence of G proteins? Similarly, can CD69/ S1PR1 complex (either monomer or dimer) accommodate ß-arrestin? I am not suggesting the provision of actual data, but based on the structures, modeling can suggest whether steric hindrance can permit such a complex.

Yes, we were able to purify a CD69-S1PR1 (2:2) complex without G proteins and determined its structure at a relatively low resolution (4.7 Å) (see Author response image 1). When superimposed to both S1PR1s in the CD69-S1PR1 (2:2) complex, the two β-arrestin 1 molecules from the NTSR1-βarrestin 1 complex (PMID 31945771) not only clash with each other but also collide with the other S1PR1 (Author response image 1). However, β-arrestin 1 rotates ~60° in the M2R-β-arrestin 1 structure (PMID 31945772) compared to that of NTSR1-βarrestin 1. With this orientation, the two protomers are compatible with each other when aligned to S1PR1 (Author response image 1). It will be interesting in future work to determine if the CD69-S1PR1 2:2 complex exists in cells and to test if it interacts with β-arrestins.

**Author response image 1. sa2fig1:** (**A**) Cryo-EM structure of the 2:2 CD69-S1PR1 complex. (**B**) Superimposition of CD69-S1PR1 with NTSR1-β-arrestin 1 complex (6UP7). Clashes are indicated by dashed ovals. (**C**) Superimposition of CD69-S1PR1 with M2R-β-arrestin 1 complex (PDB: 6U1N).

Reviewer #2 (Recommendations for the Authors):This paper by Chen et al. reports the functional characterization of a membrane protein, CD69, as an agonist of the S1p1 receptor to induce receptor activation and internalization without lipid ligands. The authors further determined a cryo-electron microscopy (cryo-EM) structure of the human S1p1-Gi complex with CD69. The structure revealed how CD69 activates S1p1 by interacting with the transmembrane helix 4 (TM4) of S1p1. The structural findings are further validated by mutagenesis studies. The results provide a novel mechanism for the modulation of GPCR signaling by other membrane proteins.The findings and conclusions of this paper are well supported by data. The cryo-EM structure determination looks solid. All functional assays were performed with appropriate replicates and controls. I would suggest addressing or clarifying the following issues:1) Does CD69 just function as an agonist of S1p1 in vivo? What are other possible functional roles of CD69 involving its extracellular region? Does CD69 need to form a homodimer to activate S1p1?

As discussed in the manuscript, CD69 upregulation in lymphocytes is associated with rapid loss of S1PR1 function in promoting egress from tissues into blood or lymph. Prior work has shown that S1P gradients are critical for lymphocyte egress to occur. If S1P abundance is greatly increased in tissues (due to inhibition of its metabolism), lymphocyte egress is blocked. We suggest that by activating S1RPR1 in cis, CD69 causes lymphocytes to lose their ability to sense the egress promoting S1P gradient, thereby promoting their retention in tissues.

We did not detect any evidence for the CD69 ectodomain being involved in the interaction with S1PR1, consistent with earlier biochemical findings (PMID 20463015). Other work has suggested the CD69 ectodomain may interact with Galectin-1 and this may modulate T cell cytokine production; an interaction with the S100A8/S100A9 complex has also been suggested (reviewed in PMID 28475283). A further study provided evidence for CD69 interaction with extracellular myosin light chain (Myl) 9 and Myl12 to regulate T cell function in the inflamed lung (PMID 28783682). It will be of interest in future studies to understand whether molecules that interact with CD69 in trans can influence the internalization or signaling by the S1PR1-CD69 complex.

Like related C-type lectins, CD69 is believed to form a disulfide linked homodimer during synthesis in the endoplasmic reticulum. Therefore, while a monomer is predicted to be able to engage S1PR1, we don’t consider it likely that a monomeric form would be physiologically relevant.

2) The Gi-induced and β-arrestin-independent receptor internalization is interesting. What could be the molecular mechanism? Gi protein itself doesn't internalize upon activation.

We don’t yet know the mechanism. We note in the manuscript that GRK2 and dynamin have been shown to participate in S1PR1 internalization in response to S1P, and that these factors promote internalization of some receptors independently of β-arrestins. Future studies will need to test whether these or other molecules are involved.